# A Research on Durability Degradation of Mineral Admixture Concrete

**DOI:** 10.3390/ma14071752

**Published:** 2021-04-02

**Authors:** Xupeng Chen, Zhuowen Sun, Jianyong Pang

**Affiliations:** 1School of Civil Engineering and Architecture, Anhui University of Science and Engineering, Huainan 232001, China; jypang@aust.edu.cn; 2School of Transportation and Civil Engineering, Nantong University, Nantong 226019, China

**Keywords:** concrete, mineral admixture, salt erosion, grey relational analysis, micro analysis

## Abstract

In order to study the degradation laws and mechanisms of admixture concretes with single-added SO_4_^2−^ and composite of Mg^2+^ and SO_4_^2−^, respectively, the durability tests were conducted on three types of mineral admixture concretes (concretes with single-added metakaolin (MK), single-added ultra-fine fly ash (UFA), and composite of metakaolin and ultra-fine fly ash (MF), and one reference concrete. In these tests, the 10% Na_2_SO_4_ solution and the 10% MgSO_4_ solution were used as the erosion medium, and the drying-wetting circle method was applied. It can be seen from the compressive tests and grey relational analysis that the MK admixture can improve the anti-Na_2_SO_4_-erosion capability of the concrete significantly, but weaken its anti-MgSO_4_-erosion capability; the UFA admixture can improve both the anti-Na_2_SO_4_-erosion and the anti-MgSO_4_-erosion capability of the concrete; and the composite admixture has superimposed effects and can enhance erosion resistance against these two erosion mediums. The phase composition and the changes of the macro morphology and the micro structure during the erosion process caused by mono sulfate ions and complex ions has been observed through X-ray diffraction (XRD), FTIR spectrum (FTIR), and scanning electron microscope (SEM), based on which it was determined that the erosion of single-added SO_4_^2−^ ions can produce erosive outputs of ettringite, gypsum, and mirabilite in the concrete, and cause corner scaling or deformation. Mg^2+^ and SO_4_^2−^ reacted in the concrete and produced brucite, M-S-H, ettringite, and gypsum, etc. The erosion of complex ions can cause scaling of the cement mortar and aggregate from the surface or the desquamation of corners.

## 1. Introduction

Concrete might be eroded by multiple ions in saline soil or salty lake environments, such as SO_4_^2−^, HCO_3_^−^, CO_3_^2−^, Mg^2+^, Cl^−^, Ca^2+^, K^+^, Na^+^ [1,2], among which SO_4_^2−^ and Mg^2+^ have the greatest impact on plain concrete, reducing its durability significantly [3,4,5]. The reason why sulfate ions cause the durability degradation of common concrete is mainly because it produces Ettringite (AFt) in the capillary of slurry, which further creates crystallization pressure, leading to concrete cracks [6]. For magnesium ions, its decalcification effect on Calcium silicate hydrate (C-S-H) gel can replace the Ca^2+^ in C-S-H gel, changing C-S-H into Magnesium silicate hydrate (M-S-H) with no cementitious character [7]. Generally, the sulfate attack on concrete can be classified into two categories: (a) physical corrosion, and (b) chemical corrosion. Physical corrosion causes the concrete degradation via salt crystals in the supersaturated solution in the pores. On the contrary, chemical corrosion is characterized by expansion and cracking caused by the production of ettringite and gypsum. Liu et al. [8] have conducted a number of studies on the sulfate attack on concrete in semi-immersed conditions, based on which they determined, by microstructure analysis, that fly ash speeds the deterioration of concrete, mostly in a chemical way instead of a physical way.

Although a number of studies [9,10,11,12,13] have been done on the sulfate resistance of concrete, they only adopted single sulfate as the erosion medium in most cases, and applied the immersed or semi-immersed method in all the experiments. In fact, many erosion ions co-work to erode the concrete simultaneously which produce quite different results [14,15]. According to the existing literature, in saline soil or salty lake regions, because of the critical climate and temperature differences, drying-wetting cycle environment changes can be seen frequently, which impacts greatly different from only the immersion condition [16,17]. In Western China there are many salt lakes and saline areas rich in sulfate ions and magnesium ions. The construction, underground pipes, and bridges in these areas suffer sulfate attack significantly. Some water conservancy systems are also affected by erosion of both sulfate and magnesium ions, especially in the upper and middle reaches of the Yellow River. In sea water, since the sulfate and magnesium ions have the greatest impacts on the concrete, the cross-sea bridges, the subaqueous tunnels, the offshore drilling platforms, harbors and coastal engineering also face the erosion damages. As for railways, the Chengdu-Kunming Railway, the Jiaozhou-Liuzhou railway, the Wuhan-Guangzhou passenger railway, and the Shijiazhuang-Taiyuan passenger railway suffer from the corrosion as well. In the salt lake regions in Western China, because of the critical climate and temperature difference, the environment there changes from dry to wet alternatively. Additionally, the underground water level, the reservoir water level, and the seawater level also change all the time with the dry-wet environment changing conditions. As a matter of fact, for the concrete exposed in such an alternating dry-wet environment, the migration and pore absorption effects are the key factors resulting in concrete damage [18]. The dry-wet alternation enhances the expansion of salt crystallization and the sulfate attack promotes the production of ettringite [19]. As a result, the concrete cracks are formed [20]. The damage of dry-wet alternation to the concrete is much greater than that of immersion. In salt lake areas and water level changing areas the concrete degrades much faster than those in other areas. Therefore, it is necessary to combine the dry-wet alternation and sulfate erosion to study the physical (dry-wet alternation) and chemical (salt erosion) effects on concrete, so as to find ways to improve the durability of concrete.

There are many ways to improve the concrete performance to resist sulfate attack, Both coating [21,22] and a low water-glue ratio can improve the concrete resistance against chemical and physical damage. In addition, the adding of an air-entraining agent and compacting agent can reduce the crystallization pressure effectively and prevent the physical damage caused by the salt crystallization and expansion [23,24]. With urbanization and industrialization, the concrete consumption increases rapidly year-by-year [25]. To produce one ton of cement, it costs a large amount of petroleum and coal resources, and exhausts 0.8–1.2 tons of CO_2_ [26,27]. The CO_2_ exhausted in cement production accounts for 5–8% in all the greenhouse gases. Using a mineral admixture instead of cement can not only reduce the greenhouse gas exhaustion so as to protect the environment, but also improve the concrete durability [28,29,30]. Studies conducted by Wang Qicai [31] show that the admixture of fly ash and slag powder admixture in concrete can improve the sulfate erosion resistance. Studies conducted by Du Jianmin [32] show that the 20% fly ash in concrete can improve the sulfate erosion resistance significantly in half-immersing environment. Sideris [33] uses two kinds of natural cinerites and finds that, in most cases, the cinerite can improve the sulfate erosion resistance.

In these papers, the traditional mineral admixtures, like common fly ash or mineral slag powder admixture, are used, which would have a negative impact on the mechanical properties of the concrete in the early stage. Highly active mineral admixtures, such as MK and UFA, can improve not only the concrete compressive strength in the early stage, but also the durability after compounding. MK [34,35] is active in catalysis and crystallization and can work as filler. It can promote the production of C-S-H gel and optimize the micro-structure of cement mortar so to improve the compactness and the mechanical properties of concrete. Goncalves [36] has tested MK’s impact on the concrete resistance to erosion of magnesium sulfate. The substitution ratios are set at 10% and 20% while the concentration of magnesium sulfate solution is 5%. The result shows that the higher the substitution ratio, the more severe the damage caused to the MK concrete in the magnesium sulfate solution. The results of Mardanic tests [37] show that when the substitution ratio of MK is set at 10%, the concrete degradation caused by expansion in the magnesium sulfate solution can be reduced significantly. Little research has been done to compare the roles of UFA and MK in concrete resistance performance over sulfate salt, and composite salt, therefore there’s still no conclusion yet. A great deal of work has been done on the erosion mechanism of magnesium salt and sulfate, but there are no general rules explaining the changes of the internal phase composition, macro morphology, and microstructure of high active admixture concrete eroded by mono ions or complex ions in different stages of cycling. For this reason, a 10% Na_2_SO_4_ solution and 10% MgSO_4_ solution were taken as the corrosive medium, the drying-wetting circle method was used as the study method, and the compressive strength and strength loss rate were used as the indicators to study the durability degradation law of admixture concrete in this study. The grey relational analysis was applied to analyze the durability of admixture concrete, which were respectively added with mono MK, mono UFA, and composite MK and UFA. Ten different kinds of concretes were used in this experiment, and the micro analysis has been carried out respectively on ordinary concrete [38], MK concrete [20], and UFA concrete [39]. which experienced sulfate erosion. The results show that, the erosion products in MK concrete and UFA concrete are different, in concentrations, but the same in type. Therefore, M10F30 concrete is selected as the research object for micro analysis in this study. The M10F30 concrete, which is one of the innovative points in this paper, mixes two different active admixtures into the concrete, presenting complicated micro structure with good research significance. Compared with other groups of concrete, M10F30 concrete shows better performance, and greater application prospects in practical engineering. Therefore, X-ray diffraction, Fourier infrared spectroscopy, and scanning electron microscopy are also used to analyze the changes and laws of internal phase composition and degradation of M10F30 concrete during the dry-wet circle. It provides a theoretical basis for predicting the durability of concrete with admixture in the western region of China.

## 2. Experiment Overview

### 2.1. Materials of the Experiment

Metakaolin is made by mineral kaolin after being roasted at 500~900 °C. MK used in this study is made by Shanghai Lingdong Co., Ltd. (Shanghai, China), particle size: 1.8 μm. The particle size distribution is shown in Figure 1. As shown in Figure 2a and Figure 3a, it is white powder with irregular shape.

UFA is made by Henan Zhengzhou Huifeng New Material Co., Ltd. (Zhengzhou, China), particle size: 5.95 μm. As shown in Figure 2b and Figure 3b, it is a yellowish-brown powder with round shape, and greater average particle size than MK from a micro perspective.

The common PO42.5(a Chinese standard ordinary Portland cement) cement with density of 3.13 g/cm^3^ and the specific surface area of 3320 cm^2^/g was applied in this study. The mixture proportion of admixture concrete is shown in Table 1.

Medium sand from Huaihe river is used as the fine aggregate with a fineness modulus of 2.9. As for the coarse aggregates, gravels with continuous grading of 5~15 mm were used. The polycarboxylate superplasticizer made of Shanxi Qinfen Construction Materials Co., Ltd. (Weinan, China) was applied in this study, achieving a 37% water reduction rate.

### 2.2. Proportion of Admixture Concrete

The proportion of mineral admixture concrete is as shown in Table 2. The poured concrete was put into molds, which was then put onto the vibrostand for making it more dense. After that, the material was put in standard curing room at 20 ± 2 °C, 95% relative humidity for 24 h. Finally, the concretes were demolded and returned to the curing room for 28 days (d), and were taken out on the 28th day for testing.

### 2.3. Experimental Method

Non-standard 100 mm × 100 mm × 100 mm cubic specimens were taken in this experiment, for which the uniaxial compression test is carried out according to Standard of GB/T 50081—2002 [40]. During the test, the press should be set with stable and uniform speed. The rate of loading should be about 3 mm/min. Since non-standard specimens were used in this study, the conversion coefficient must be multiplied during calculation. The reduction coefficient of the compressive strength of the cube is 0.95, while the same data of the splitting tensile strength is 0.85.

According to GB/T 176—2008 [41], the optimized barium sulfate precipitation method (titration) is used to check the content of sulfate ions in the concrete powder. An impact drill is used to drill and get core powder of concrete blocks at different stages of erosion. Nine holes were drilled symmetrically in each block and mixed the powder drilled from depth of 0, 5, 10, 15, 20 mm together, which should be ground by mortar and tested later. The sampling process is shown in Figure 4.

The dry state and wet state are as shown in Figure 5a,b. The dry-wet cycle process is briefly described as follows: immersing for 15 h → drying in room temperature 1 h → drying in oven at 60 °C 7 h → cooling in room temperature for 1 h. Based on the GB/T 50082-2009 [42], we also referred to the research of Wang [38] and Zhao [43], and the chemical composition of ions in the soil and groundwater in Western China. A 10% solution of Na_2_SO_4_ and a 10% solution of MgSO_4_ are used as the erosion medium. The experiment includes four periods, i.e., the dry-wet alternation for 20, 40, 80, and 120 cycles respectively. During the test, the PH value of the immersing solution was checked once every seven days, and the PH value was kept at 7.00 ± 0.50 with pure concentrated sulfuric acid. The solution was changed every 20 days.

After each period, tests were made to check the changes of compressive strength of each set of concrete. At the same time, the compressive strength of common concrete in water is tested for comparison. Finally, micro observation was performed with XRD and SEM.

## 3. Experiment Results and Analysis

### 3.1. Analysis of the Laws of Sulfate Ion Transport

Although the erosion solutions in this study were all 10% sulfate solutions, they varied in molar concentration of sulfate ions. In order to determine the sulfate iron concentrations in different circulation periods from the surface to the internal part of all groups of concretes, the titration method was used. It can be seen from Figure 6 that the content of sulfate ions on the surface of the concrete blocks in the Na_2_SO_4_ solution is slightly lower than that in the MgSO_4_ solution. With the pores going deeper, the difference between sulfate ion content in the Na_2_SO_4_ solution and the MgSO_4_ solution was enlarged. It is mainly the magnesium ions that make the difference, since the magnesium ions react with calcium hydroxide earlier than sulfate ions and produces new layer of brucite, which accelerate the emerging of micro cracks. Furthermore, the concrete pH value decreases, which makes the C-H-S in the ettringite degrade and turn into M-S-H. The increase of M-S-H speeds up the forming of gypsum, thereby changing the concrete components and their proportions, degrading the physical property consequently, and enlarging the thickness loss. For concrete in the sodium sulfate solution, sodium ions react in the concrete and produce mirabilite in most cases. From the macro view, this product is the crystallized salt attached to the surface of the cracks of the concrete, having little impact to the concrete properties. In the sodium sulfate solution, the ettringite and gypsum produced by sulfate ions in the concrete play a very important role in the erosion.

### 3.2. Impacts of Na_2_SO_4_ and MgSO_4_ Erosion on the Compressive Strength of MK Admixture Concrete

The strength loss formula is given as below in order to figure out the impacts of different proportions of admixture on concrete strength:(1)$$\Delta f_{\alpha }=\frac{f_{N}-f_{0}}{f_{0}}$$
where $\Delta f_{\alpha }$ is the strength loss ratio, $f_{N}$ is the compressive strength after cycle for *N* times, and $f_{0}$ is the compressive strength of concrete specimens experiencing no circulation.

As shown in Figure 7a, the compressive strength changes of single-added admixture concrete in 10% Na_2_SO_4_ solution can be divided into four stages: ascending, flat, descending, and quick descending stages. In the early stage, due to the great activity and smaller particle size of MK compared to cement, the powerful pozzolanic effect and filling effect make the concrete with MK more dense than common concrete, and the reduction of pores makes migration of the erosion solution even more difficult, so the concrete with mono MK additive shows a rising strength tendency in both ascending and flat stages at 0–40 cycles compared to the reference concrete. At the 40th–120th cycles, the strength of all groups decreased. This is mainly because a large amount of ettringite and gypsum were produced, which induced expansion and crack of admixture concrete, causing its strength to decline.

Figure 7b shows the compressive strength changes of admixture concrete in 10% MgSO_4_ solution_._ It can be divided into three stages: ascending, descending, and quick descending. Since MK contains a large amount of active Al_2_O_3_ and SiO_2_, it can react with CH in cement and produce C-S-H cementing material so as to improve the concrete porosity. However, Mg^2+^ can react with C-S-H and obtains hydration products, such as M-S-H and M-C-S-H, which will soften the cement basement and cause the degradation of the compressive strength of the concrete [44]. The side product of gypsum causes concrete expansion and property degradation as well. The MK admixture can weaken the MgSO_4_ erosion resistance of the concrete significantly. Compared to the reference concrete, the compressive strength of the MK admixture concrete is improved more slowly after 0–20 circulations, but decreased more rapidly after 20–120 circulations.

Figure 8 shows the loss rate of concrete compressive strength with different proportions of MK admixture. It is clear that when the concrete is eroded by Na_2_SO_4_, the higher proportion of the MK admixture, the less loss of compressive strength. The maximum loss rate of compressive strength is only 1.45~30.02%. When the concrete is eroded by MgSO_4_, the higher proportions of MK admixture, the more loss of compressive strength. The maximum loss rate of compressive strength varies from 41.65~52.46%. The erosion effects of MgSO_4_ solution on MK admixture concrete is much greater than that of the Na_2_SO_4_ solution.

### 3.3. Impact on Compressive Strength of UFA Admixture Concrete after the Erosion Corrosion of Na_2_SO_4_ and MgSO_4_

It can be seen from Figure 9 sthat the UFA concrete shows different tendencies of compressive strength changes in different solutions. In 10% Na_2_SO_4_, solution, the changes can be divided into four stages: ascending, flat, descending, and quick descending. In 10% MgSO_4_ solution, the changes can be divided into three stages: ascending, descending, and quick descending. After 0–20 cycles, the concrete compressive strength keeps rising in both solutions because, on one hand, the cementing materials like the cement are still in the hydration stage while, on the other hand, a small amount of SO_4_^2−^ ions get in and react with the hydration products in the slurry and produce a slightly corrosive medium, which works just like the filler. After 20–120 cycles, the concrete compressive strength degrades. Compared to the reference concrete, the compressive strength of the UFA admixture concrete degrades slowly in two types of solutions. The main reason is that UFA reacts with Ca(OH)_2_ in cement, which reduces the production of ettringite or gypsum, reducing the progress of the expansion and cracks caused by ettringite and gypsum.

Figure 10 shows that. With the increase of UFA proportion, the compressive strength loss rate of the concrete in two solutions reduce gradually. The main reason of compressive strength degradation is that the sulfate salt reacts with Ca(OH)_2_ and produce gypsum (CaSO_4_·2H_2_O). The expansion of gypsum causes the compressive strength degradation. Ettringite is another factor that causes the loss of compressive strength. Ettringite can be produced in the concrete curing process. Additionally, in the reaction process between gypsum (CaSO_4_·2H_2_O) and tricalcium aluminate (C_3_A), secondary ettringite can also be produced [6]: C_3_A + 3CSH_2_ + 26H→C_6_AS_3_H_32_. With the increase of the UFA proportion, the contents of Ca(OH)_2_ and gypsum can be consequently reduced. Since UFA replaces part of the cement, the content of C_3_A can be reduced due to the higher proportion of UFA. Therefore, the higher proportion of UFA, the better performance of the concrete in resisting sulfate attack. As shown in Figure 7a,b, the maximum loss rate of the compressive strength of UFA admixture concrete in the 10% solution of Na_2_SO_4_ is 5.67~26.55%. In the 10% solution of MgSO_4_ it is 5.52~34.2%. Both are lower than that of reference concrete. Definitely UFA helps to improve the anti-Na_2_SO_4_ and -MgSO_4_ erosion performance of concrete.

### 3.4. Impact on MK and UFA Compressive Strength of Composite Admixture Concrete after the Erosion of Na_2_SO_4_ and MgSO_4_

As shown in Figure 11a, after the Na_2_SO_4_ erosion and 0–40 cycles, the compressive strength of the reference concrete changes from an ascending to a descending tendency. The composite admixture containing MK and UFA with continuous grading cement particles produces superimposed effects, and further reduces the pores in the cementing system, achieving better concrete durability. Therefore, the compressive strength of composite admixture concrete keeps rising at this time. After 40–80 cycles, the compressive strength of the composite admixture concrete degrades more slowly compared to that of the reference concrete. Figure 11b shows that when the proportion of MK is set unchanged, the higher proportion of UFA, the less loss of compressive strength.

In the MgSO_4_ solution, Mg^2+^ in MK plays a very important role in the erosion to the composite admixture concrete. The compressive strength changes of the composite admixture concrete is similar to that of the reference concrete. After 0–20 cycles, the compressive strength changes from an ascending to a descending tendency. As shown in Figure 12b, the higher proportion of UFA, the less loss of compressive strength. Viewing from Figure 8, Figure 10 and Figure 12, it is clear that the reference concrete, the mono admixture concrete, and the composite admixture concrete have more loss of compressive strength in the 10% MgSO_4_ solution than in the 10% Na_2_SO_4_ solution_._ Therefore, the MgSO_4_ solution has stronger erosion effects on mineral admixture concrete than the Na_2_SO_4_ solution.

### 3.5. Grey Relational Analysis

Grey relational degree is often used to analyze the relations among different factors in the grey system [45,46]. The analysis method can be used to determine the impacts of different affecting factors on matters, and select optimal solutions based on the calculation results. Therefore, in order to determine how different admixtures and different proportions would affect the concrete performance in Na_2_SO_4_ and MgSO_4_ erosion resistance, the Deng’s grey relational analysis method is used here [47]. The compressive strength of concrete cured in water after each cycle period is set in the main sequence, marked as *x*_0_. The compressive strength of the reference concrete, MK admixture, UFA admixture, and composite admixture concretes eroded by Na_2_SO_4_ and MgSO_4_ solutions after each cycle period are set in the influence factor sequence, marked as *x*_1_~*x*_9_. The calculation is made as shown below.

1)Calculate the original value of each sequence, and use the initial value method to undimensionalize the original test data and turn it into a sequence convenient for comparison:(2)$$x_{i}(k)=\frac{x_{i}^{\prime }(k)}{x_{i}^{\prime }(1)}(i=0,1,2........10;k=1,2,3,4,5),$$
where $x_{i}(k)$ refers to the data after undimensionalization, $x_{i}^{\prime }(k)$ refers to the original data, and *x*_*i*_′ (1) refers to the first original test data. The detailed original strength sequence and sequence after undimensionalization are listed in Table 3.2)Set up the difference transformation matrix

Calculate the absolute value of difference between the main sequence and the influence factor sequence and set up the difference transformation matrix [48] ∆_1_ and ∆_2_.
$$\Delta_{1}=\left[\begin{array}{ccccc} 0 & 0.017 & 0.153 & 0.405 & 0.600\\ 0 & 0.033 & 0.053 & 0.227 & 0.496\\ 0 & 0.015 & 0.006 & 0.098 & 0.270\\ 0 & 0.014 & 0.007 & 0.070 & 0.211\\ 0 & 0.028 & 0.216 & 0.326 & 0.538\\ 0 & 0.077 & 0.032 & 0.086 & 0.320\\ 0 & 0.119 & 0.025 & 0.013 & 0.251\\ 0 & 0.028 & 0.021 & 0.084 & 0.263\\ 0 & 0.015 & 0.010 & 0.080 & 0.213\\ 0 & 0.046 & 0.074 & 0.006 & 0.071 \end{array}\right]\Delta_{2}=\left[\begin{array}{ccccc} 0 & 0.060 & 0.089 & 0.285 & 0.544\\ 0 & 0.021 & 0.173 & 0.320 & 0.548\\ 0 & 0.016 & 0.209 & 0.364 & 0.596\\ 0 & 0.008 & 0.244 & 0.451 & 0.656\\ 0 & 0.113 & 0.049 & 0.199 & 0.397\\ 0 & 0.123 & 0.022 & 0.122 & 0.359\\ 0 & 0.164 & 0.105 & 0.059 & 0.188\\ 0 & 0.009 & 0.128 & 0.345 & 0.548\\ 0 & 0.051 & 0.074 & 0.218 & 0.453\\ 0 & 0.032 & 0.079 & 0.194 & 0.420 \end{array}\right]$$

3)Calculate the grey relational degree

Correlation index:$r(x_{0}(k),x_{i}(k))=\frac{\min_{i}\min_{k}\Delta_{0i}(k)+\zeta \max_{i}\max_{k}\Delta_{0i}(k)}{\Delta_{0i}(k)+\zeta \max_{i}\max_{k}\Delta_{0i}(k)}$, ζ refers to the resolution ratio, usually taking the general value of 0.5, which mainly aims to reduce the impact of high absolute value to the significance results.
(3)$$\begin{array}{c} r_{1}(x_{0},x_{i})=\left[\begin{array}{cccccccccc} 0.673 & 0.739 & 0.841 & 0.866 & 0.667 & 0.792 & 0.828 & 0.833 & 0.859 & 0.891 \end{array}\right](i=1\sim 10)\\ r_{2}(x_{0},x_{i})=\left[\begin{array}{cccccccccc} 0.708 & 0.695 & 0.679 & 0.661 & 0.738 & 0.774 & 0.781 & 0.711 & 0.741 & 0.757 \end{array}\right](i=1\sim 10) \end{array}$$
where $r_{1}(x_{0},x_{i})$ refers to the correlation degrees of concretes in all test groups in the 10% Na_2_SO_4,_ solution and $r_{2}(x_{0},x_{i})$ refers to the correlation degree of concretes in all test groups in the 10% MgSO_4_ solution_._ Comparing the correlation degrees of the test groups in the two erosion solutions, it can be seen that when concrete is eroded by the Na_2_SO_4_ solution, the correlation degree of each admixture concrete group is in this order: composite admixture concrete > MK admixture concrete > UFA admixture concrete > reference concrete. The composite admixture concrete has the best Na_2_SO_4_ corrosion resistance performance, then comes the MK admixture concrete, followed by the UFA admixture concrete, and the reference concrete is the worst. The higher proportion of mineral admixture, the higher correlation degree, which means the MK, UFA, and the composite of MK and UFA can slow the compressive strength degradation when the concrete is eroded by SO_4_^2^^−^ ions. When the proportion of MK is 10% and the proportion of UFA is 30%, the concrete has the best Na_2_SO_4_ erosion resistance performance. All these results are in line with the experiment conclusions above.

When the concrete is eroded by MgSO_4_ solution, the erosion resistance performances of admixture concrete are in this order: UFA admixture concrete > composite admixture concrete > reference concrete > MK admixture concrete. The higher proportion of MK, the lower correlation degree, which shows that MK plays a negative role in the SO_4_^2−^ and Mg^2+^ erosion resistance performance of admixture concrete. When UFA is added to concrete, the higher proportion of UFA, the higher correlation degree. This shows that UFA can improve the SO_4_^2−^ and Mg^2+^ erosion resistance performance. The admixture concrete show the best MgSO_4_ erosion resistance when the proportion of UFA is 30%. All these results are in line with the experiment conclusions above.

### 3.6. Macro Analysis

Figure 13 is the macro view of concrete M10F30 in Na_2_SO_4_ solution. After 20 cycles, there are some small defects on the surface of concrete. After 40 cycles, concrete scaling and small cracks emerged because of the internal expansion caused by ettringite and gypsum. After 120 cycles, the concrete expansion and scaling appear, showing O-shaped deformation.

Figure 14 shows the concrete M10F30 in the MgSO_4_ solution. After 20 cycles, there are more pores on the surface and surface scaling can be seen. After 40 cycles, the cement aggregate peels off on many parts of surface. The internal cement aggregate can be observed by eyes. After 120 cycles, the internal cement mortar structure is damaged seriously. Large blocks fall off both on the cross-section and on the sides.

### 3.7. Phase Composition Analysis

#### 3.7.1. X-ray Diffraction Analysis

Micro analysis is conducted on the M10F30 concrete by XRD and SEM. Figure 15a shows the XRD spectrum of admixture concrete after erosion of 10% Na_2_SO_4_ solution_._ It shows a significant diffraction peak of Ca(OH)_2_, and a small diffraction peak of ettringite at the time having no erosion. Since concrete has aggregates, the diffraction peaks of quartz and calcite are also significant. In addition, there is the diffraction peak of albite, which is mainly formed by N-A-S-H losing water at the late stage of hydration [38]. With the erosion by Na_2_SO_4_ solution, the diffraction peak of Ca(OH)_2_ becomes weak while the diffraction peak of ettringite(Aft) gets intensified, because the SO_4_^2−^ ions get in by diffusion and convection, thus reacting with Ca(OH)_2_ in the concrete and produce ettringite. After some dry-wet alternations, the diffraction peaks of CaSO_4_ and Na_2_SO_4_ can be seen. Moreover, the diffraction peak of CaSO_4_ gets intensified more and more with the increase of cycles. After 80 cycles, the concrete has some cracks and more Na_2_SO_4_ solution gets in, and crystals separate. Now the diffraction peak of Na_2_SO_4_ becomes intense and that of mirabilite can been seen. After 120 cycles, the diffraction peak of mirabilite becomes intense. The cracks on the surface of admixture concrete get larger and large blocks of cement mortar and aggregate peel off. The compressive strength degrades significantly.

Figure 15b shows the XRD spectrum of admixture concrete eroded by 10% MgSO_4_ solution, which is quite different from that of Na_2_SO_4_. After 20 cycles, the diffraction peaks of brucite, MgSO_4_, gypsum, and CaSO_4_ appear and the diffraction peaks of gypsum and CaSO_4_ are comparatively low at this time because the number of cycles is not large enough. With the increase of cycle number, the diffraction peak of ettringite becomes weaker and those of gypsum and CaSO_4_ become more intense. That is because most of the Ca(OH)_2_ in the concrete has been consumed and the pH value declines. The production of CaSO_4_ is fastened and CaSO_4_ begins to change into gypsum at that time. After even more cycles, the diffraction peaks of gypsum and brucite become more intense. The ettringite and C-S-H undergo a decalcification reaction and the production of brucite and CaSO_4_ is fastened. After decalcification, C-S-H changes into M-S-H, which causes the concrete property degradation [49]. After 120 cycles, the diffraction peak of gypsum comes to the top. The concrete expands and cracks and the compressive strength degrades further.

#### 3.7.2. FTIR Analysis

M-S-H needs to be confirmed through FTIR. Figure 16 shows the Fourier infrared spectrum of M10F30 group concrete (with 10% MK and 30% UFA) and reference concrete after experiencing 120 cycles in 10% MgSO_4_ solution. Three characteristic absorption peaks of O-H can be seen at 3648.5, 3452.7, and 168.81 cm^−1^ in test samples. These absorption peaks are relatively small, indicating that most calcium hydroxide has been consumed after 120 cycles. The characteristic absorption vibration peaks of C–O can been seen at 881.5 and 779.2 cm^−1^, which correspond to the erosion products shown in the figure above. The characteristic absorption peaks come from quartz and calcite. There are stretching vibration absorption peaks of S–O at 1161.2 and 608 cm^−1^ and absorption peaks of AlO_6_ at 552.3 and 862.5 cm^−1^, therefore, it can be known that there must be the erosion product of ettringite in the concrete. The pH value in the concrete must have declined after 120 cycles. In this condition, C-S-H reacts with MgSO_4_ and produces M-S-H. when the pH value in the concrete is comparatively low, there are characteristic absorption peaks of M-S-H at 1000 and 1040 cm^−1^ [50], and the characteristic absorption peak values show that the M-S-H content in M10F30 concrete is much more than that in the reference concrete.

### 3.8. SEM Analysis

Figure 17 is the SEM of the admixture concrete eroded by Na_2_SO_4_. After 20 cycles, the concrete obtains a comparatively denser internal structure. Due to the pozzolanic effect of MK, certain hydration products occur, like the cube-shaped (C-A-H) and (C-S-A-H). A small amount of ettringite crystals get between the pores in the concrete. After 40 cycles, most calcium hydroxide has been consumed and the pH value of the concrete declines. As shown in Figure 17b, there are small hexagonal calcium hydroxide crystals. Some large and small ettringites get between the cementing material, just like needles. Some pores can been seen as well. It shows that at the early stage of the cycle that the Na_2_SO_4_ erosion to the concrete mainly causes the expansion of ettringite. After 120 cycles, as shown in Figure 17c, the gypsum sheets become large and form layer upon layer in the concrete. Large blocks of cement mortar and aggregate fall. Large pores can bee seen. The properties of the concrete degrade significantly.

Figure 18 shows the SEM photos of admixture concrete eroded by the MgSO_4_ solution. Figure 18a shows that when the cement is in hydration, the spherical UFA particles work as fillers. After 20 cycles, the compressive strength of concrete ascends. There are a large number of fibrous brucite and a small ettringite crystals in brucite layer. After 40 cycles, as shown in Figure 18b, there are a large number of dense, needle-like ettringite and a small, stumpy gypsum crystals. Meanwhile, at this time, cracks appear in the inner part of the specimen, which provides channels for MgSO_4_ erosion and provides space for erosion outputs superimposing. Compared with the diffraction peak and the SEM photos of Na_2_SO_4_, MgSO_4_ would induce gypsum earlier during the process of corroding admixture concrete, and have a larger amount of small ettringite. This is mainly because M-S-H can soften and damage the cement, allowing SO_4_^2−^ to get into the concrete easier and cause the secondary destruction of the admixture concrete in the corrosive solution. Most ettringite crystals are produced after the secondary destruction. After 120 cycles, as shown in Figure 18c, the brucite and gypsum on the surface of the concrete grow gradually and turn into sheets of brucite and gypsum. Meanwhile, pores appear in the concrete, which greatly reduces the performance of the admixture concrete.

## 4. Conclusions

In order to compare the erosion impacts of different dosages of two different active mineral admixtures on the concrete performance of sulfate erosion resistance, compressive strength tests were conducted on concretes with different cycle age; the titration method was applied to determine the sulfate ions clearly; and grey relational analysis was used to process the compressive strength data. In addition, M10F30 is selected as the research object because of the following reasons: (1) Though the quantities of main erosion products in each group of concrete blocks are different, they are of the same types; (2) though their micro structures change at different times, they have similar changing rules; (3) M10F30 has better performance than that of other groups of concrete blocks; and (4) M10F30 mixes two mineral admixtures of MK and UFA at the same time, which is representative in micro analysis. Meanwhile, XRD and FTIR spectrum are adopted to analyze and compare the macro and micro morphology changes of concretes in various cycles, from which conclusions are obtained as below:The addition of MK can improve the Na_2_SO_4_ erosion resistance of concrete. On the other hand, due to the effects of Mg^2+^, the addition of MK can reduce the MgSO_4_ erosion resistance. UFA can improve the Na_2_SO_4_ and MgSO_4_ erosion resistance of the concrete significantly. UFA and MK composite admixture concrete has the best Na_2_SO_4_ erosion resistance performance, but the weaker MgSO_4_ erosion resistance than mono UFA admixture concrete.Through grey relational analysis, it can be determined that the concrete has the best Na_2_SO_4_ erosion resistance performance when the proportion of MK is 10% and the proportion of UFA is 30%. UFA admixture concrete has the best MgSO_4_ erosion resistance performance, especially when the proportion of UFA is 30%.Through XDR and FTIR, it is known that the erosion products made when sulfate salt corrodes the admixture concrete include ettringite, gypsum, and mirabilite. The erosion products made when complex ions corrode the admixture concrete include ettringite, gypsum, brucite, and M-S-H.Through macro and SEM observation, it can be determined that the destruction of the concrete made by mono sulfate salt can be divided into three stages: production of ettringite, erosion of gypsum and ettringite, and growth of ettringite and gypsum crystals. The destruction made by complex ions can be divided into three stages: initial production of brucite and ettringite, co-destruction from M-S-H, ettringite, and gypsum, and growth of crystals of brucite, ettringite, and gypsum.

The highly active mineral admixture of MK and UFA can not only improve the sulfate resistance of concrete, but also reduce the concrete cost and protect the environment. Magnesium ions play a very important role in the MK concrete anti-erosion performance. If the magnesium ion content in the environment is comparatively high, the proportion of MK must be eliminated, or other highly active admixtures should be mixed with MK to further improve the sulfate resistance of concrete and reduce the harms brought by magnesium ion invasion.

Admixture concrete eroded by either mono sulfate or complex ions would degrade in three stages. The experiments show that after the second stage, the concrete degrades rapidly, with much more erosion products. Some erosion resistance measures must be done in this stage so as to slow the concrete degradation.

## Figures and Tables

**Figure 1 materials-14-01752-f001:**
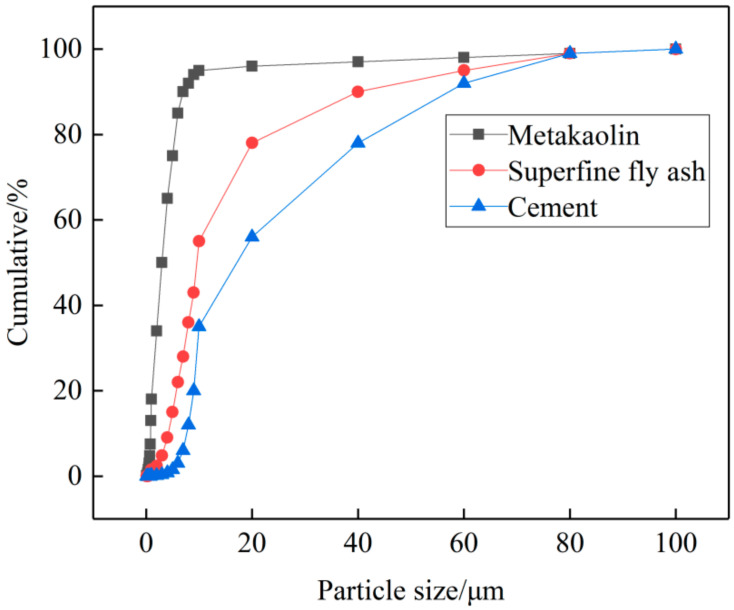
Particle size distribution of raw materials.

**Figure 2 materials-14-01752-f002:**
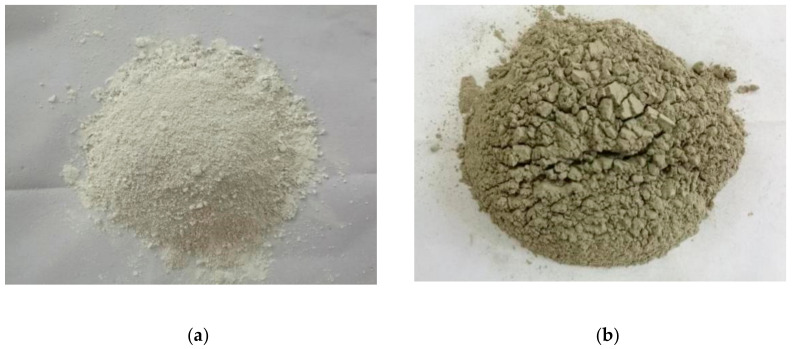
Macro morphology of mineral admixtures. (**a**) Metakaolin, (**b**) Superfine fly ash.

**Figure 3 materials-14-01752-f003:**
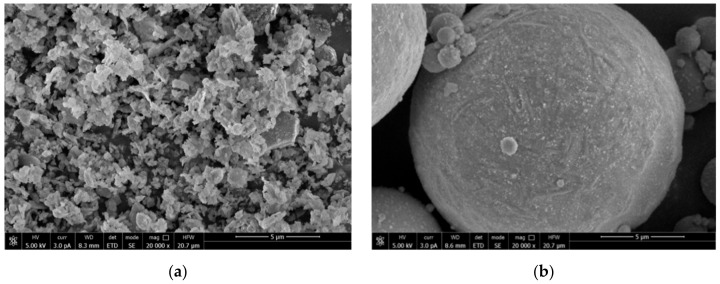
Micro morphology of mineral admixtures. (**a**) Metakaolin, (**b**) Superfine fly ash.

**Figure 4 materials-14-01752-f004:**
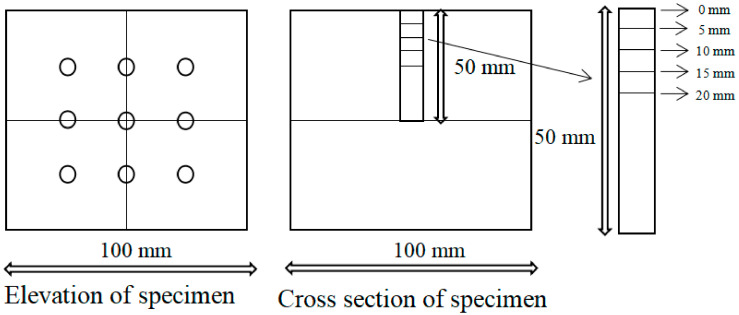
Sampling process diagram.

**Figure 5 materials-14-01752-f005:**
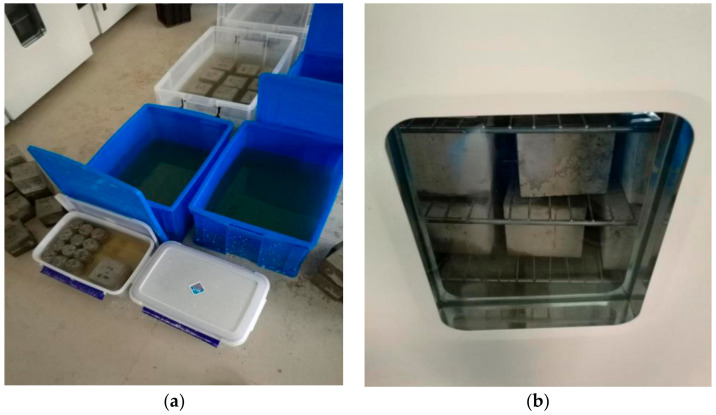
Test specimens during soaking and drying. (**a**) wet state, (**b**) dry state.

**Figure 6 materials-14-01752-f006:**
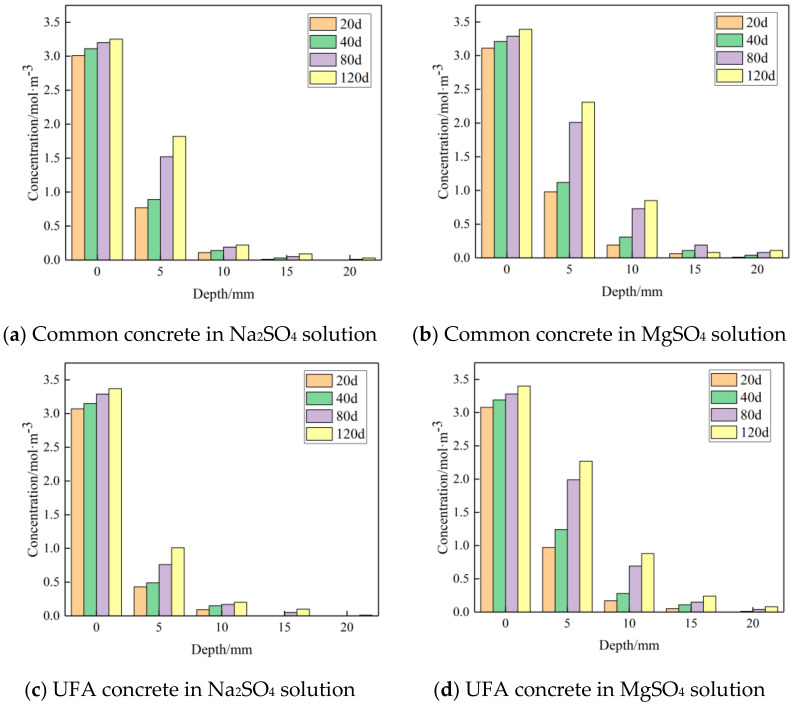
Sulfate ion test values at different erosion cycles and depths.

**Figure 7 materials-14-01752-f007:**
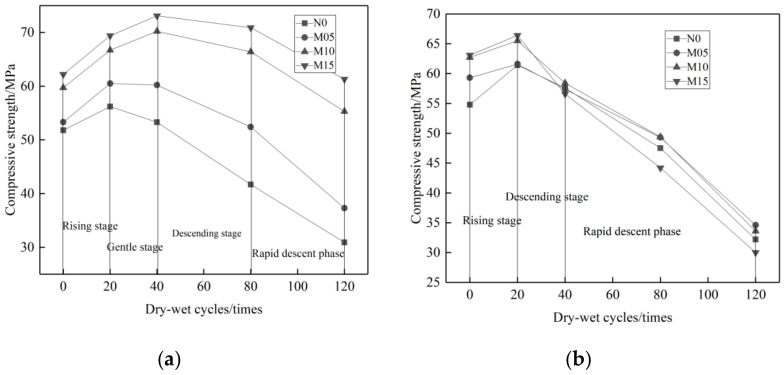
Variation curve of compressive strength of MK concrete with erosion times under different erosion conditions. (**a**) 10% Na_2_SO_4_, (**b**) 10% MgSO_4_.

**Figure 8 materials-14-01752-f008:**
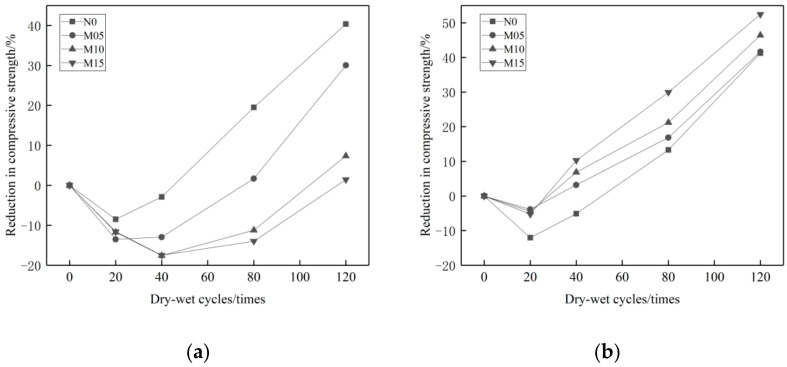
Loss rate of compressive strength of concrete with MK dosage under different erosion conditions. (**a**) 10% Na_2_SO_4_, (**b**) 10% MgSO_4_.

**Figure 9 materials-14-01752-f009:**
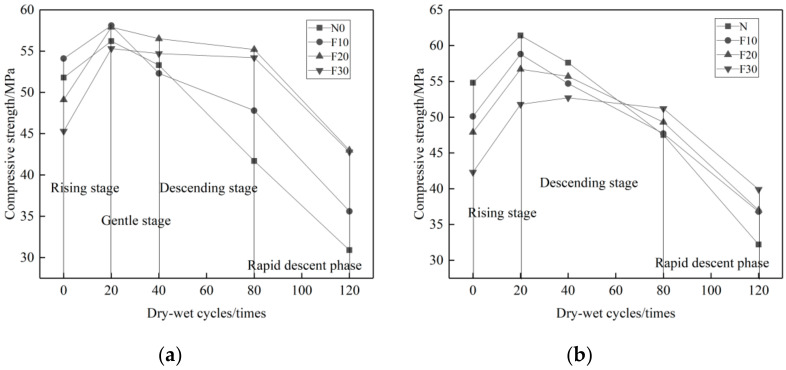
Variation curve of compressive strength of UFA concrete with different erosion times under different erosion conditions. (**a**) 10% Na_2_SO_4_, (**b**) 10% MgSO_4_.

**Figure 10 materials-14-01752-f010:**
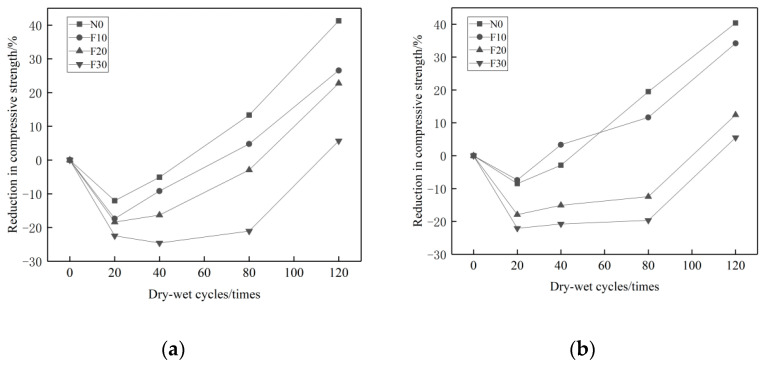
Loss rate of compressive strength of concrete with UFA dosage under different erosion conditions. (**a**) 10% Na_2_SO_4_, (**b**) 10% MgSO_4_.

**Figure 11 materials-14-01752-f011:**
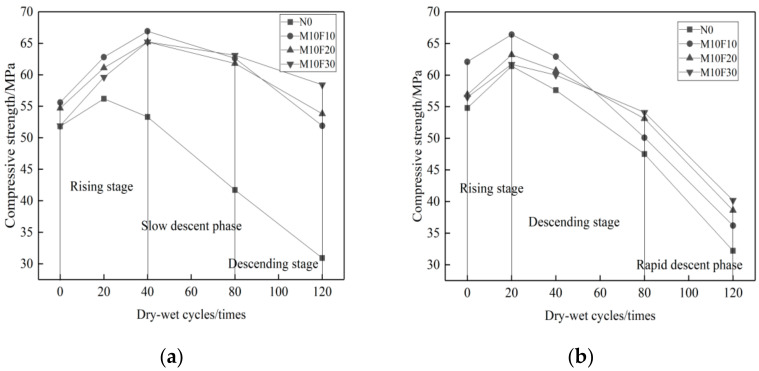
Variation curve of compressive strength of double admixtures concrete with different erosion times under different erosion conditions. (**a**) 10% Na_2_SO_4_, (**b**) 10% MgSO_4_.

**Figure 12 materials-14-01752-f012:**
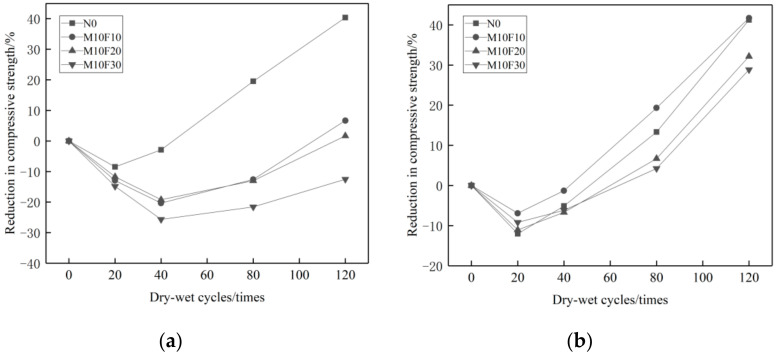
Loss rate of compressive strength of concrete with double admixtures under different erosion conditions. (**a**) 10% Na_2_SO_4_, (**b**) 10% MgSO_4_.

**Figure 13 materials-14-01752-f013:**
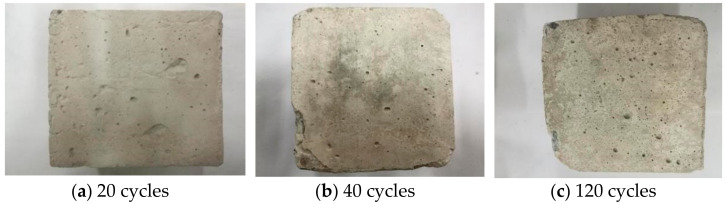
Macro morphology of concrete in the Na_2_SO_4_ solution.

**Figure 14 materials-14-01752-f014:**
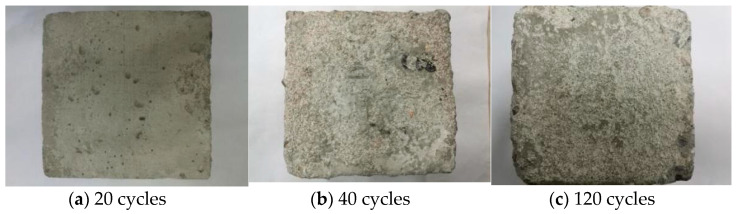
Macro morphology of concrete in the MgSO_4_ solution.

**Figure 15 materials-14-01752-f015:**
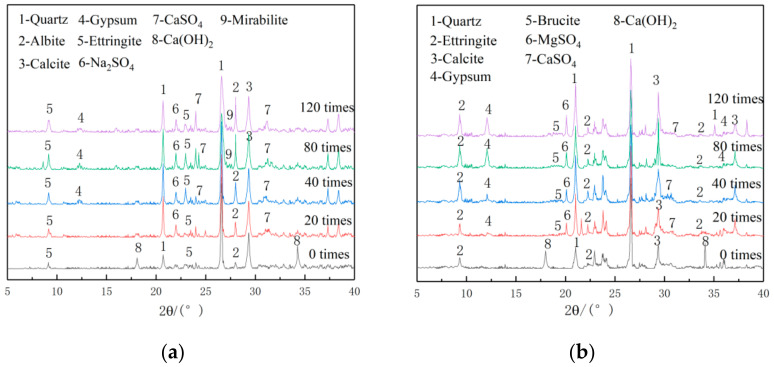
XRD spectrum of admixture concrete eroded by different erosion solutions. (**a**) 10% Na_2_SO_4_, (**b**) 10% MgSO_4_.

**Figure 16 materials-14-01752-f016:**
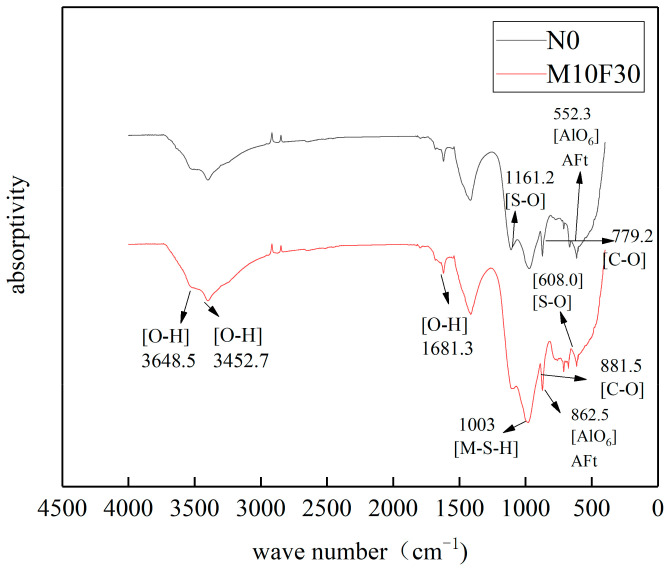
Infrared spectrum of M10F30 concrete in magnesium sulfate solution.

**Figure 17 materials-14-01752-f017:**
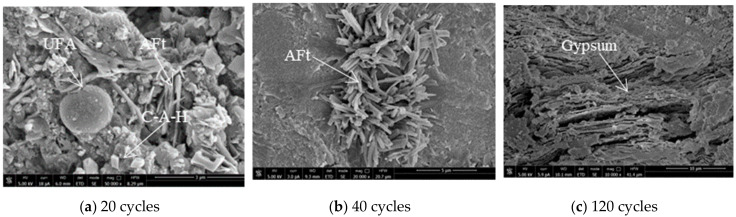
SEM photo of Na_2_SO_4_ solution erosion of M10F30 concrete.

**Figure 18 materials-14-01752-f018:**
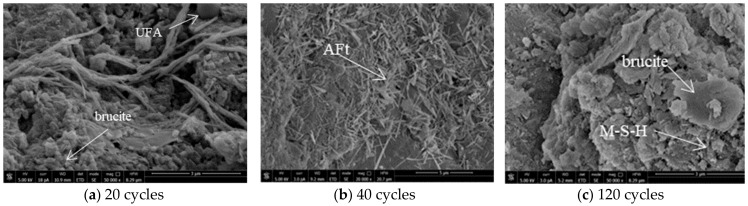
SEM photos of MgSO_4_ solution erosion of M10F30 concrete.

**Table 1 materials-14-01752-t001:** Chemical composition of cementitious materials.

Composition	CaO	Al_2_O_3_	SiO_2_	Fe_2_O_3_	MgO	Na_2_O	K_2_O	SO_3_	P_2_O_5_	ZrO_5_
Cement	68.12	3.31	21.8	2.78	0.75	0.04	0.68	1.03	0.31	0.32
MK	0.03	41.21	56.41	0.62	0.07	0.12	0.19	0.20	0.11	0.26
UFA	2.21	35.61	54.64	5.18	0.41	0.31	0.45	0.51	0.21	0.19

**Table 2 materials-14-01752-t002:** Mix proportion of admixture concrete.

Admixture Content	NO	Cement kg/m^3^	MK kg/m^3^	UFA kg/m^3^	Sand kg/m^3^	Aggregate kg/m^3^	Water kg/m^3^	Water Reducing Agent kg/m^3^
Normal	N0	500	0	0	619	1101	180	3
M5%	M05	475	25	0	619	1101	180	4
M10%	M10	450	50	0	619	1101	180	4
M15%	M15	425	75	0	619	1101	180	4
F10%	F10	450	0	50	619	1101	180	3
F20%	F20	400	0	100	619	1101	180	3
F30%	F30	350	0	150	619	1101	180	3
M10%; F10%	M10F10	400	50	50	619	1101	180	4
M10%; F20%	M10F20	350	50	100	619	1101	180	4
M10%; F30%	M10F30	300	50	150	619	1101	180	4

**Table 3 materials-14-01752-t003:** Original test data and dimension elimination data of concrete under erosion.

Number	10%NaSO_4_ Erosion					10%MgSO_4_ Erosion				
	Curing for 28 Days	Cycle for 20 Days	Cycle for 40 Days	Cycle for 80 Days	Cycle for 120 Days	Curing for 28 Days	Cycle for 20 Days	Cycle for 40 Days	Cycle for 80 Days	Cycle for 120 Days
*x*′_0_	51	56.2	60.3	61.7	61	54.8	58.1	62.5	63.1	62
*x* _0_	1	1.10	1.18	1.21	1.20	1	1.06	1.14	1.15	1.13
*x*′_1_	51.8	56.2	53.3	41.7	30.9	54.8	61.4	57.6	47.5	32.2
*x* _1_	1	1.08	1.03	0.81	0.60	1	1.12	1.05	0.87	0.59
*x*′_2_	53.3	60.5	60.2	52.4	37.3	59.3	61.6	57.4	49.3	34.6
*x* _2_	1	1.14	1.13	0.98	0.70	1	1.04	0.97	0.83	0.58
*x*′_3_	59.7	66.7	70.2	66.4	55.3	62.7	65.5	58.4	49.4	33.6
*x* _3_	1	1.12	1.18	1.11	0.93	1	1.04	0.93	0.79	0.54
*x*′_4_	62.2	69.4	73.1	70.9	61.3	63.1	66.4	56.6	44.2	30.0
*x* _4_	1	1.12	1.18	1.14	0.99	1	1.05	0.90	0.70	0.48
*x*′_5_	54.1	58.1	52.3	47.8	35.6	50.1	58.8	54.7	47.7	36.8
*x* _5_	1	1.07	0.97	0.88	0.66	1	1.17	1.09	0.95	0.73
*x*′_6_	49.1	57.9	56.5	55.2	43	47.9	56.7	55.7	49.3	37.0
*x* _6_	1	1.18	1.15	1.12	0.88	1	1.18	1.16	1.03	0.77
*x*′_7_	45.3	55.3	54.7	54.2	42.8	42.3	51.8	52.7	51.2	39.9
*x* _7_	1	1.22	1.21	1.20	0.94	1	1.22	1.25	1.21	0.94
*x*′_8_	55.6	62.8	66.9	62.6	51.9	62.1	66.4	62.9	50.1	36.2
*x* _8_	1	1.13	1.20	1.13	0.93	1	1.07	1.01	0.81	0.58
*x*′_9_	54.7	61.1	65.2	61.8	53.8	56.9	63.2	60.7	53.1	38.6
*x* _9_	1	1.12	1.19	1.13	0.98	1	1.11	1.07	0.93	0.68
*x*′_10_	51.9	59.6	65.2	63.1	58.4	56.9	63.2	60.7	53.1	38.6
*x* _10_	1	1.15	1.26	1.22	1.13	1	1.09	1.06	0.96	0.71

P.S. *x*′_0_ is the original compressive strength data of common concrete in standard curing condition after each period of cycle.

## Data Availability

Data is contained within the article.

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
