# Peer review of "A Research on Durability Degradation of Mineral Admixture Concrete"

_materials, 2021, doi:10.3390/ma14071752_

Round 1

Reviewer 1 Report

Present manuscript is dealing with the resistance of concrete incorporating mineral additives to aggresive environment. The performed work is based on the scalling tests and accompanying analytical procedures. I have following commend and recommendation:

  • the title is strange, that should be corrected,
  • introduction is too simple, I recommend to carried out thorough review of the topic,
  • authors should update the literature, there is referred lot of quite old articles; in addition, archive of this journal should be checked,
  • binder composition should be revised, sum of components in Table 1 is far from 100%,
  • some abbreviation are presented before their explanation,
  • I miss discussion of obtained results on the basis of those published works, that is the main weakness of the paper. "Results" part contains predominantly empirical evaluation,
  • formal aspects of the manuscript should be improved.

I recommend major revision.

Author Response

Thank you very much for your comments and suggestions!

all my Modification instructions and the revised article are written in the attached document.

Reviewer 2 Report

All my comments and suggestions are written in the attached document.

Author Response

(The authors gave the same response as above.)

Reviewer 3 Report

The manuscript is focused on evaluating the sulphate attack on the concretes with mineral admixtures. The authors studied a potential positive role of metakaolin and fly ash incorporated into concrete composites. The topic of the paper is very interesting and the authors present original experimental results. In my opinion, however, it is necessary to clarify several issues in the experimental part and results.

  1. I am not an English native speaker, however I think, there is a need for an English revision of the paper.
  2. I am sorry, but I am afraid, the title of the paper is not correct and not clear. In addition, the term “compound” is not suitable there. Do the authors mean “mutual” or “conjunctive” impact?
  3. Abstract – all abbreviations should be explained in the text where they first appear. Please explain the UFA, MK abbreviation given in the Abstract.
  4. Please check the chemical formulas, if the charge of an ion is equal to one, no number is usually given, only minus sign.
  5. Please correct the references e.g. [1,2] in the text, now it seems confusing, like being charge of the sodium cation.
  6. The reference No. 16 is missing in the Introduction.
  7. Please check the terminology in the manuscript. Do authors mean water-to-binder ratio when mentioned the “water-glue ratio”?
  8. Table 2 – the units of the particular components given in the table are missing. Please add the units.
  9. The explanation is needed regarding the selection of the corrosive sulphate solutions. Why were not chosen the solutions with the same concentration of sulfate ions?Although both solutions are of 10%, the content of sulfate ions in them is not the same.
  10. The authors stated in the manuscript: “From figure 5, figure 7 and figure 9, it’s clear that the benchmark concrete, the mono admixture concrete and the compound admixture concrete have more loss of mechanical property in 10% solution of MgSO4than in 10% solution of Na2SO4. Hence theMgSO4 solution has stronger corrosion to mineral admixture concrete than the Na2SO4 solution.” Is this conclusion not expected in advance as the concentration of sulphate ions in MgSO4 is higher than in Na2SO4?
  11. In connection to the previous comments, how can the authors evaluate the Mg2+ and Na+ role in corrosion process if the sulphate concentrations are not equal in the corrosive media? Authors should point to this fact in methodology and when discussing the results.

Author Response

(The authors gave the same response as above.)

Reviewer 4 Report

English language quality is not good enough not only in the aspect of grammar  but also in the field of terminology (e.g.: water-glue ratio; mineral slag powder admixture; anti-sulfate corrosion performance;, half-soaking environment and many others) - the extensive professional edition by native speaker and specialist in the field of concrete technology is indispensable

The title of the paper is not clear: the term "Compound Corrosive Ion’s"  is not precise - it would be better to denominate the type of corrosion in the title . Prefered term for such a typ aof concrete compound is additive not admixture (according to European Standards)

The experiment is not fully described: 

the materials used - waht do you mean by statement "Particle size: 1.8μm" It couldn`t be true - grain size distribution curve is recommended. What was the production process? what raw material was used for this additives.

shape and dimensions of the specimens, number of the specimens in the series, conditions in “curing room” (chmaber?), testing methods for compressive strength (standards?), description of soaking-drying cycles not specify the duration of stages. What about the concentrations of solutions during exposure - was it constant? did you monitor it? Or did you change the solutions after each set of cycles?

the term "Grey relational analysis" seems to be more frequently used.

And generally - what are the new findings due to described experiment - please specify this as the aim of the paper and aphasize this in the conclusions

The conclusions are only detailed - some general effects of your research should be expressed also, maybe some recommendations about the application od tested materials?

Author Response

(The authors gave the same response as above.)

Round 2

Reviewer 1 Report

The manuscript was corrected in accordance with my previous remarks. The manuscript could be published as it is, however I recommend to modify caption of Fig. 17 a Fig. 18 during proofreading; 20 times, 40 times and 120 time evoke the maginification applied during SEM analysis, to modify to "20 cycles", e.g. would be more suitable, in my opinion.

Author Response

Thank you very much for taking the precious time to review my article for the second time!
Please see the attachment.
all my Modification instructions and the revised article are written in the attached document.

Reviewer 2 Report

The manuscript has been corrected significantly. However, it still contains some flaws which should authors correct. All my comments and suggestions are written in the attached document.

Author Response

(The authors gave the same response as above.)

Reviewer 3 Report

The article was very badly checked for me, as the authors did not provide a version of the manuscript with the marked changes, nor did they mark the exact lines and parts that were revised in the document answering to the reviewer. 

In my opinion, the manuscript needs a serious English check, there are many typos in the article (resistnce etc.). I also recommend checking the text by an English speaking chemist, there are several not common used terms, e.g. "compound ion" ? I think the term is not correct.

Besides that, there are still several formal shortcomings in the text:

Abstract - In my opinion, it is enough to explain the abbreviations in the text as follows: description (abbreviation), e.g. metakaolin (MK); ultra-fine-ash (UFA), etc.

Abstract - not all abbreviations are still explained, e.g. abbreviations of the analytical methods XRD, FTIR, SEM.

Table 2 - some units are given in capital letters Kg, other not kg.

Author Response

I am very sorry to bring inconvenience to your review and thank you very much for taking the precious time to review my article for the second time!
Please see the attachment.
all my Modification instructions and the revised article are written in the attached document.

Reviewer 4 Report

The scientific quality of the paper is now much higher. The chcaracteristics of used microfillers are now sufficient, the general aim of the research is clearly visible and  discussed generally in the conclusions. Missed details of testing procedures are also included in the text.

In my opinion the paper could be accepted for the publication in Material

Author Response

Thank you very much for your affirmation of my article! I am very excited about it!

Thank you again for spending your precious time reviewing and evaluating for me!Through this revision, the quality of my article is much better.And I also learned a lot from your comments and opinions.